# Serum Anti-Spike Antibodies Are Not Affected by Immunosuppressants in SARS-CoV-2 Vaccinations Given to Brazilian Patients with Inflammatory Bowel Disease

**DOI:** 10.3390/healthcare11202767

**Published:** 2023-10-19

**Authors:** Magno Luís Costa Pereira, Jessica Pronestino de Lima Moreira, Luís Cristóvão Porto, Vania Maria Almeida de Souza, Beatriz Cunta Gonçalves, Amanda de Barros Sampaio, Matheus Figueiredo Moutela, Larissa dos Reis Farha, Bárbara Cathalá Esberard, Renata Fernandes de Amorim, Heitor Siffert Pereira de Souza, Ana Teresa Pugas Carvalho

**Affiliations:** 1Inflammatory Bowel Disease Outpatients Unit, Piquet Carneiro Polyclinic, Rio de Janeiro State University, Rio de Janeiro 20950-003, Brazil; mlcosta16@gmail.com (M.L.C.P.); larissa.farha@gmail.com (L.d.R.F.); besberard@outlook.com (B.C.E.); amorim.rf@gmail.com (R.F.d.A.); atpugas@yahoo.com.br (A.T.P.C.); 2Department of Bromatology, Faculty of Pharmacy, Fluminense Federal University, Niterói 24241-002, Brazil; jpronestino@id.uff.br; 3Clinical Pathology Service, Piquet Carneiro University Polyclinic, Rio de Janeiro State University, Rio de Janeiro 20950-003, Brazil; lcporto@uerj.br (L.C.P.); vania.almeida73@gmail.com (V.M.A.d.S.); 4Division of Gastroenterology, Pedro Ernesto University Hospital, Rio de Janeiro State University, Rio de Janeiro 20551-030, Brazil; beatrizcunnta@gmail.com (B.C.G.); amandadbsampaio@gmail.com (A.d.B.S.); matheusmoutela13@gmail.com (M.F.M.); 5Department of Clinical Medicine, Federal University of Rio de Janeiro, Rio de Janeiro 21941-913, Brazil; 6D’Or Institute for Research and Education (IDOR), Rua Diniz Cordeiro 30, Botafogo, Rio de Janeiro 22281-100, Brazil

**Keywords:** inflammatory bowel disease, COVID-19, immunosuppressants, vaccine, immune response, antibody titers

## Abstract

This study aimed to evaluate humoral responses after vaccination against severe acute respiratory syndrome coronavirus 2 (SARS-CoV-2) of patients with inflammatory bowel disease (IBD). Patients with IBD enrolled in a tertiary outpatient unit were followed up between September 2021 and September 2022 via serial blood collection. Immunoglobulin G antibody titers against SARS-CoV-2 were measured before administration and 1 and 6 months after the administration of two doses of different vaccination regimens. The results were compared with those of a healthy control group obtained during the same period. The mean pre-vaccination antibody titers were 452.0 and 93.3 AU/mL in the IBD (*n* = 42) and control (*n* = 89) groups, respectively. After two doses of the vaccine, the titers significantly increased in both groups (IBD, 8568.0 AU/mL; control, 7471.0 AU/mL; *p* < 0.001). One month after the second dose, no significant differences were observed between the two groups (*p* = 0.955). Significant differences between vaccination schemes in the IBD group were observed, with higher titers in those who received Pfizer, younger patients (*p* < 0.005), and those with a previous coronavirus disease 2019 (COVID-19) infection (*p* < 0.012). The use of immunosuppressants and immunobiologicals did not affect the overall humoral response to COVID-19 vaccine in patients with IBD, but specific vaccine regimens, age, and previous coronavirus infection significantly did. This study reinforces the positive impact of booster doses and the safety of SARS-CoV-2 vaccination.

## 1. Introduction

The coronavirus pandemic peaked in April 2021, with Brazil having the second highest number of deaths and the third highest number of cases worldwide [1,2]. Currently, over 80% of Brazilians (170 million people) have received at least two vaccine doses [3]. Considering the negative numbers related to the pandemic, it is crucial to prioritize vaccination and boosters for high-risk populations, particularly those with compromised immune systems, such as patients with inflammatory bowel disease (IBD) receiving immunosuppressive or immunobiological therapy [4,5]. Recently, a fifth dose of the vaccine has been considered for this group, as the humoral response to vaccines may be lower than that of the general population. 

Crohn’s disease (CD) and ulcerative colitis (UC), the two major forms of IBD, are complex inflammatory disorders whose incidences are progressively increasing worldwide [6,7]. Although the etiopathogenesis of IBD is not completely understood, a large body of evidence indicates underlying defects in innate and adaptive immunity [8,9] and abnormal immune reactivity against gut commensal microorganisms [10]. Because of its chronic nature, patients often require continuous treatment, including immunosuppressants, which can have serious side effects [11]. However, the humoral response to COVID-19 vaccines and the need for vaccine boosters in this group of patients using immunosuppressants or immunobiotics are still not fully established [12,13]. Therefore, it is of paramount importance to understand the pandemic pattern and role of vaccination in this population, which involves the risk of progression to severe disease and the humoral response achieved and maintained after complete immunization with different vaccines [14,15].

The introduction of SARS-CoV-2 vaccines has led to dramatic reductions in hospitalization and death [15]. Previous studies on the effectiveness of COVID-19 vaccination have focused on responses to commonly administered vaccines worldwide, especially adenovirus and mRNA vaccines, such as BioNTech Pfizer, Moderna, and AstraZeneca [16,17]. Recent studies on IBD patients have shown that post-vaccination adverse events are similar to those in the general population, with no increased risk of disease exacerbation [18,19]. However, the use of immunosuppressants in patients with IBD may lead to variable responses to vaccines against other diseases [20,21,22,23], making the effectiveness of COVID-19 vaccination in IBD patients, as well as the required level of immune response for protection, questionable. Hence, this prospective study aimed to evaluate the humoral responses to different types of vaccines against SARS-CoV-2 in patients with IBD and compare them with those in a healthy population.

## 2. Methods

### 2.1. Study Design and Selection of Patients

This prospective, single-center, observational, longitudinal study enrolled patients who were followed up regularly at the Inflammatory Bowel Disease Outpatient Unit, Piquet Carneiro Polyclinic of the State University of Rio de Janeiro, a tertiary referral center, between September 2021 and September 2022. Eligible individuals were patients aged 15–75 years with a diagnosis of IBD (CD or UC), supported by routine clinical, endoscopic, histological, and imaging parameters. Patients who had not been vaccinated against COVID-19 were consecutively selected to participate regardless of the current therapy for IBD. Convenience sampling was conducted during the study period. A total of 31 patients with CD, 11 with UC, and 89 controls without comorbidities were included. Those with concomitant autoimmune diseases and/or HIV/AIDS, individuals who did not sign the informed consent form, pregnant women, those who refused to vaccinate, those with unclassified IBD, those in the postoperative period of less than 6 months or with total colectomy, those with evidence of abdominal abscess or colonic mucosal dysplasia, and those with cancer or acute or chronic enteric infection (e.g., *Clostridioides difficile*) were excluded.

### 2.2. Study Protocol and Procedures

Between September 2021 and September 2022, blood samples were collected from all individuals enrolled in the study at three different time points: before vaccination and 1 month and 6 months after the second dose of the vaccine. The vaccines analyzed in this study were AstraZeneca (a non-replicating viral vector), BioNTech Pfizer (mRNA), and CoronaVac (an inactivated virus). Samples were collected at the IBD outpatient unit in a reserved room after vital signs were checked by the nursing team. After collection, the samples were registered in the MV Seoul system using patient registration data. After identification, the samples were centrifuged and stored at −20 °C until the date of shipment (within 1 week) to the UNADIG-Fiocruz/RJ Diagnostic Center.

Titers of immunoglobulin G (IgG) antibodies against the SARS-CoV-2 spike receptor binding domain were determined using a chemiluminescent microparticle immunoassay. In this assay, the spike protein was derived from the wild-type virus before the emergence of variants. For the qualitative (N, against the nucleoprotein) and quantitative (S, against the S1 subunit of the receptor binding domain of the SARS-CoV-2 spike protein) determination of IgG antibodies, we ran an SARS-CoV-2 IgG automated immunoassay (for N) and SARS-CoV-2 IgG II Quant assay (for S) (Abbott Laboratories, Abbott Park, IL, USA), using the Architect i2000sr platform (Abbott) and according to the manufacturer’s recommendations. The assay uses paramagnetic microparticles coated with a nucleoprotein or the S1 subunit of the receptor binding domain of the spike protein. The response (in relative light units) was based on the IgG II standard/calibrator estimates, reflecting the quantity of IgG antibodies present. Overall, the assay exhibited 99.37% sensitivity and 99.5% specificity. Qualitative results were considered positive when the N nucleoprotein index was ≥1.4. Seropositivity was defined as ≥50 arbitrary units (AU).

The results were registered in the MV Soul system, signed, and made available in the electronic records of the patients or on the MV website. All data, including adverse events collected over 1 year of follow-up, were stored on the Google Chrome digital platform in a password-protected spreadsheet format accessible only to the researcher. Additional information is provided in Appendix A. This study followed the Strengthening the Reporting of Observational Studies in Epidemiology (STROBE) guidelines for observational studies.

### 2.3. Statistical Analysis

Statistical analyses were performed using the Statistical Package for Social Science for Windows version 24 (IBM Corp., Armonk, NY, USA). Graphs were constructed using Prism version 9.1.2 for Windows (GraphPad Software, San Diego, CA, USA). The characteristics of the patients with IBD and the control group were summarized using descriptive statistics, with means and standard deviations for continuous variables and counts and percentages for categorical variables. Antibody titers between the two groups were compared using the Mann–Whitney U test. A pairwise Wilcoxon rank-sum test was used to compare the effect of the vaccines on antibody titers at two different time points. Similarly, the groups were compared according to the vaccination scheme. Multiple comparisons between vaccination schemes in each group were performed using the Kruskal–Wallis test. Multivariate linear regression was performed to assess the influence of variables on antibody titers, using the variation in antibody titers before vaccination and 1 month after vaccination as the outcome. All tests were two-tailed, and the significance level was set at *p* < 0.05.

## 3. Results

Ninety patients with IBD were recruited, and pre-vaccination blood samples were collected for serological assays. Those who refused follow-up (*n* = 13), those with insufficient blood volume in the first sample (*n* = 3), those who did not return for further collection without justification or because they did not respond to telephone contact (*n* = 19), and those who refused additional doses of vaccines (*n* = 13) were excluded. The final sample size was 42 (Appendix A). Antibody titers against the surface protein of SARS-CoV-2 (anti-S IgG) before vaccination and 1 month after the second dose were analyzed in 42 patients with IBD and 89 patients in the control group. Six months after the second dose, all participants were followed up for antibody measurements. Patients in the IBD group were vaccinated with the first dose between days 1 and 25 after the first blood collection (median, 7.6 days). The control group was vaccinated with the first dose 1 day and 29 days after the first collection (median, 6.1 days). The sociodemographic and clinical profiles of the participants in the IBD and control groups are presented in Table 1.

### 3.1. Prior COVID-19 Infection and Vaccination Schemes

In the IBD group, 21.4% of the patients reported a previous COVID-19 infection before vaccination. All patients were asymptomatic or exhibited only mild symptoms. The vaccination scheme with two doses of AstraZeneca, Pfizer, or CoronaVac was administered to 47.6%, 33.3%, and 19.0% of patients, respectively.

In the control group, only 16.8% of patients had COVID-19 before vaccination, and all cases were asymptomatic or mild. The vaccination scheme with two doses of AstraZeneca, Pfizer, or CoronaVac was administered to 64.0%, 16.8%, and 7.9% of patients, respectively.

### 3.2. Comparison of Antibody Titers in the IBD and Control Groups

In patients with IBD, the mean antibody titers were 452.0 AU/mL before vaccination and 8568.0 AU/mL after the two vaccine doses. In the control group, the mean antibody titers were 93.3 AU/mL before vaccination and 7470.6 AU/mL after the two vaccine doses (Figure 1). In both groups, there was a significant increase in antibody titers after administration of the two vaccine doses (*p* < 0.001). Comparison of the antibody titers at 1 month after the second dose showed no difference between the IBD and control groups (*p* > 0.999).

The humoral immune response to vaccination against the SARS-CoV-2 spike protein was evaluated by measuring pre- and post-vaccination antibody titers. The medians with interquartile ranges and individual values are shown. The analysis was performed using the Kruskal–Wallis test, in which multiple comparisons between the vaccination schemes in each group were performed using Dunn’s test and the Wilcoxon matched-pairs signed-rank test for pre- and post-vaccination results. IBD, inflammatory bowel disease; SARS-CoV-2, severe acute respiratory syndrome coronavirus 2. Circles represent control individuals and triangles represent patients with IBD.

### 3.3. Potential Association between Pre- and Post-Vaccination Antibody Titers and Vaccination Schemes

One month after the two vaccine doses, the mean antibody titers of control participants vaccinated with CoronaVac, AstraZeneca, and Pfizer were 4968.7, 8212.9, and 7484.9 AU/mL, respectively. In the IBD group, the mean antibody titers were 7471.1, 6888.8, and 11593.7 AU/mL, respectively.

Individual analysis by vaccination scheme showed that in both groups, the AstraZeneca, CoronaVac, and Pfizer vaccines significantly increased antibody titers (Table 2).

When comparing the different vaccine schemes in the control group, the Pfizer scheme was clearly superior over CoronaVac at 1 month (*p* < 0.0230) and 6 months (*p* < 0.0121) after the second dose. Nevertheless, the decrease in antibody titers observed in all vaccination schemes 6 months after the second dose was not significant. In patients with IBD, the Pfizer system also induced higher antibody titers, but the differences were not significant compared with the AstraZeneca and CoronaVac systems. Unlike the control group, patients with IBD received a booster dose. More sustained antibody titers were detected 6 months after the second dose, whereas vaccination with Pfizer continued to induce relatively higher antibody titers (Figure 2).

The humoral immune response to vaccination against the SARS-CoV-2 spike protein was evaluated by measuring pre- and post-vaccination antibody titers in control individuals (A) and patients with IBD (B). The medians with interquartile ranges and individual values are shown. The analysis was performed using the Kruskal–Wallis test, in which multiple comparisons between the vaccination schemes in each group were performed using Dunn’s test and the Wilcoxon matched-pairs signed-rank test for pre- and post-vaccination results. IBD, inflammatory bowel disease; SARS-CoV-2, severe acute respiratory syndrome coronavirus 2.

### 3.4. Potential Association between Antibody Titers and Specific Features of Patients with IBD

To estimate the relationship between the clinical and demographic characteristics of the patients and antibody titers after vaccination, we analyzed the data using linear regression. Significant individual differences were observed in relation to the vaccination scheme (*p* < 0.0001), with a negative association with older age (*p* = 0.011) and a positive association with a history of COVID-19 (*p* = 0.011) (Table 3). Multiple linear regression was used to estimate the relationship between vaccine response and variables with a greater power of association with antibody titers (Figure 3). When analyzing age in the control and IBD groups isolated, antibody titers did not differ significantly (Appendix A).

### 3.5. Potential Association between Antibody Titers and the Therapeutic Regimen for IBD

Considering the different therapies used, including with and without salicylate, immunosuppressant only (azathioprine), biological only (anti-TNF alpha), and combined therapy (anti-TNF alpha plus azathioprine), all patients responded to vaccination, with significantly raised antibody titers, regardless of the therapeutic regimen adopted. Regarding the different time points, we did not find any differences in antibody titers between the distinct therapeutic groups in the pre-vaccination analyses and at 1 month and 6 months after the second dose (Figure 4).

The humoral immune response to vaccination against the SARS-CoV-2 spike protein was evaluated by measuring pre- and post-vaccination antibody titers according to different therapeutic regimens. The medians with interquartile ranges and individual values are shown. The analysis was performed using the Kruskal–Wallis test, in which multiple comparisons between the vaccination schemes in each group were performed using Dunn’s test and the Wilcoxon matched-pairs signed-rank test for pre- and post-vaccination results. IBD, inflammatory bowel disease; SARS-CoV-2, severe acute respiratory syndrome coronavirus 2.

## 4. Discussion

In this study, the outpatient unit, from which all patients with IBD and their samples were analyzed, belonged to a tertiary referral center that received patients with different levels of complexity, most of whom used biological and/or immunomodulatory therapy. Therefore, all the patients in this study had a priority indication for vaccination. It is important to highlight the small number of Brazilian studies on COVID-19 and SARS-CoV-2 vaccinations in patients with IBD, especially those involving vaccines with limited global coverage, such as CoronaVac. Moreover, this study analyzed individuals in the transition from the pre- to post-vaccination era by collecting data before and after the availability of vaccination schemes. This study focused on the evaluation of antibody titers before and 1 month and 6 months after the administration of two standard doses of different vaccination schemes. In patients with IBD, antibody titers at different time points were analyzed according to different vaccination schemes, clinical characteristics of the patients, and outcomes. The results of the IBD group were compared with those of a control group of healthy individuals with sociodemographic characteristics similar to those of the IBD group.

As is well established in the global literature, vaccination against COVID-19 has significantly contributed to the control of the pandemic. In a retrospective study conducted in Italy, Mattiuzzi and Lippi demonstrated that vaccination significantly reduced SARS-CoV-2 infections, hospitalizations, intensive care unit stays, and deaths in the general population [24]. Likewise, in patients with CD or UC, vaccination was shown to be effective and safe, protecting them from the most severe outcomes in a similar way to the general population, as concluded by Lev-Tzion et al. from Israel [25]. However, important variations among countries regarding the types of vaccines, interval between doses, transmissibility rate, and negative outcomes of infection need to be acknowledged and require further investigation. Therefore, the paucity of local data prompted us to evaluate the pattern of antibody titers in the IBD population, emphasizing a comparison with a control group, which might help us understand the influence of medications used in the treatment of IBD on the vaccine response.

Nonetheless, the decay of antibody titers over time following SARS-CoV-2 vaccination, as reported in studies with non-immunosuppressed cohorts [26], which increases the risk of COVID-19 [27], has raised concerns related to patients with IBD, particularly regarding the frequent use of immunosuppressants. As shown in previous studies in patients with IBD, infliximab was associated with attenuated serological responses to SARS-CoV-2, which were further impaired by combination therapy with immunomodulators [28]. Similarly, the decreased durability of the humoral response to SARS-CoV-2 infection was detected among patients receiving anti-TNF therapy compared with those receiving other medications, including other biologicals, which could affect long-term immunity [29]. In another study of 370 participants using different drugs to treat IBD, the use of anti-TNF and tofacitinib was associated with lower concentrations of antibodies compared with the general population, unlike what was observed with other therapeutic regimens [30]. In contrast, the results of this study show that the antibody titers in patients with IBD were similar to those in the general population after two doses of the coronavirus vaccine, regardless of the therapy used. Our findings are consistent with those of Kappelman et al., who demonstrated a high rate of seroconversion in 95% of patients in a cohort of 317 individuals with IBD after two doses of mRNA vaccine [31]. Similarly, Melmed et al. reported seroconversion in 99% of patients with IBD after two doses, regardless of the medication used to treat IBD [32].

This study also revealed other relevant associations with humoral responses in patients with IBD. In addition to reporting seroconversion and antibody titers similar to those in the general population after two doses of the vaccine, we observed that the titers changed according to the specific characteristics of the individuals. As expected and consistent with other studies, increasing age was significantly associated with lower antibody titers, an effect that is probably attributable to immunosenescence [33]. A significant independent association between age and antibody titers was observed in the multiple linear regression model, indicating the expected negative association. Nevertheless, it is important to note that this study analyzed a cohort of patients with IBD who were younger than those in most previous studies [30,34,35]. Such differences in the average age of patients in different studies may provide data on the impact of age on the response to vaccines. In addition, among several characteristics of the IBD group, the model revealed higher antibody titer concentrations in patients who were vaccinated with two doses of the Pfizer vaccine compared to the other vaccine regimens. This finding was similar to that reported by Alexander et al., who compared mRNA- and adenovirus-based vaccines [30]. Moreover, an analysis of the model indicated significantly higher serological responses in patients previously infected with COVID-19, which may be explained by the immunological memory generated by B lymphocytes after infection, as previously proposed [36].

This study had some limitations that must be acknowledged. First, the sample size is relatively small. Many initial losses occurred, largely owing to low vaccine acceptance. As COVID-19 is still regarded as a new disease, with vaccine development still in progress, global acceptance is relatively slow, similar to what occurred at the beginning of vaccination against influenza in 2009, as analyzed by Bults et al. [37]. Although SARS-CoV-2 vaccine hesitancy is a relatively common problem worldwide, partly fueled by misinformation [38], it should not exclude local governments and health system decision-makers from their responsibilities and roles in the provision of reliable information, which is critical for the rapidity and range of vaccine coverage and protection of the population. Second, the analysis performed 6 months after the second dose was biased because of the emergence of booster doses. During this period, only the IBD group was vaccinated with booster doses, preventing a direct comparison with the control group and long-term follow-up of antibody titers. Finally, as observed in most studies with similar designs, an isolated and single-center study of antibody titers measuring IgG antibodies against the spike protein (IgG anti-S) may not accurately reflect humoral immunity and the overall immune response. Considering the complexity of adaptive and innate responses against SARS-CoV-2, interpretation of vaccine immunity must be part of the broad context of immunology [39].

The simplest way to assess humoral responses to different types of vaccines is by measuring antibodies. Additionally, because COVID-19 is still considered a new disease, each process that involves protection against its development must be considered as a fundamental contribution to our knowledge of the disease. Currently, the Ministry of Health of Brazil, in agreement with the World Health Organization, recommends a second booster dose 4 months after the first. As noted by Loubet et al., the choice of a campaign with booster doses is based on the spread of new variants over time, in addition to scientific evidence of a decline in humoral response after full vaccination, mostly with two doses [40].

## 5. Conclusions

There is still much to learn about the coronavirus pandemic, and several studies have been published on this topic. Overall, the results of this study suggest that awareness regarding booster doses in patients with IBD who are receiving immunosuppressive therapy should be maintained, as the current results indicate that a booster dose of the COVID-19 vaccine enhances the antibody response.

## Figures and Tables

**Figure 1 healthcare-11-02767-f001:**
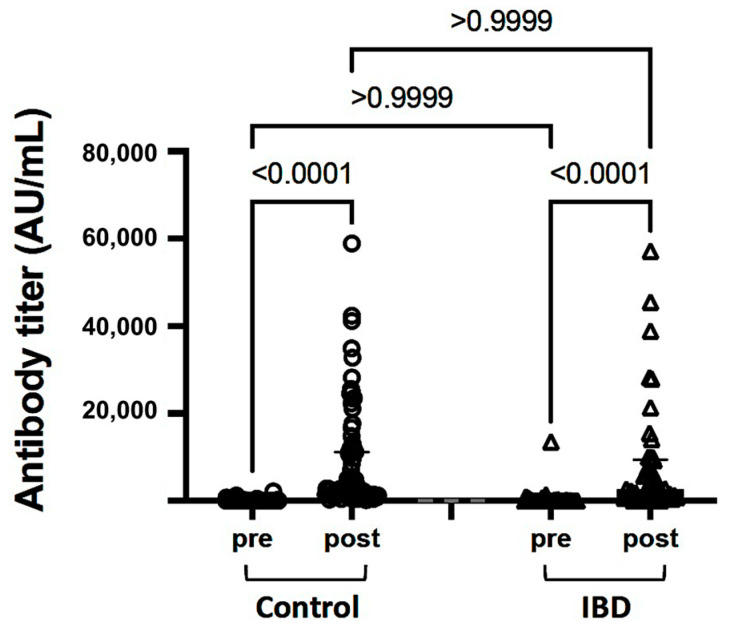
SARS-CoV-2 antibody titers in control individuals and patients with IBD.

**Figure 2 healthcare-11-02767-f002:**
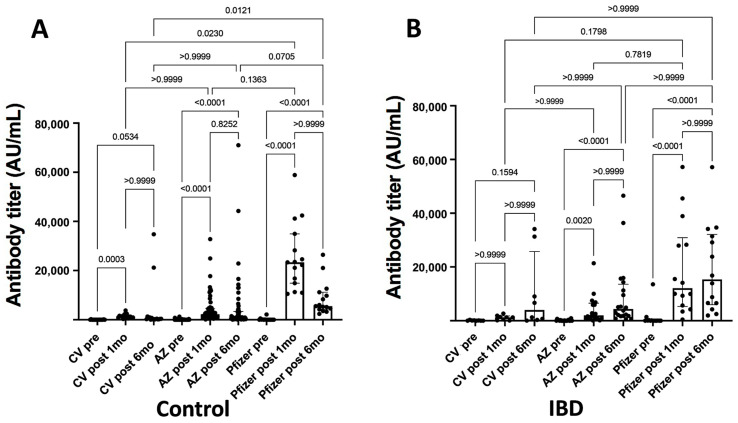
SARS-CoV-2 antibody titers in response to different vaccination schemes.

**Figure 3 healthcare-11-02767-f003:**
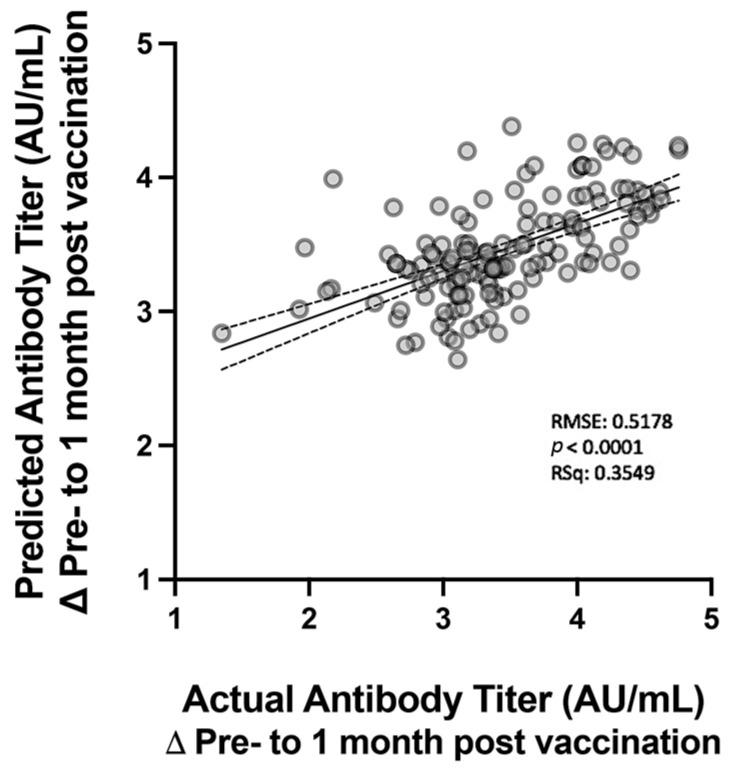
Multiple linear regression of predicted SARS-CoV-2 antibody titers modeled based on the combined rates of age, previous history of natural COVID-19 infection, and vaccination scheme. Log10-transformed individual values are shown as medians and 95% confidence intervals (lines). RSq, R-squared; RMSE, root mean square error; SARS-CoV-2, severe acute respiratory syndrome coronavirus 2; COVID-19, coronavirus disease 2019.

**Figure 4 healthcare-11-02767-f004:**
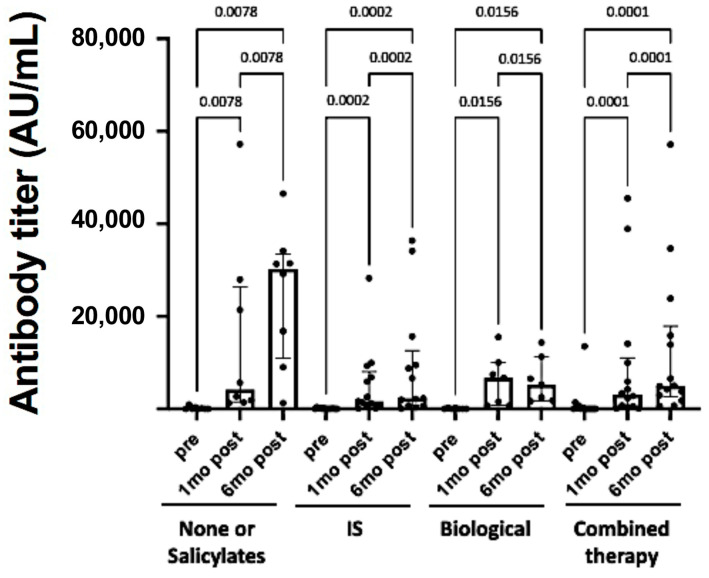
Effect of the therapeutic regimen for IBD in SARS-CoV-2 antibody titers in response to vaccination. IS, immunosuppressant (azathioprine).

**Table 1 healthcare-11-02767-t001:** Clinical and demographic characteristics of participants.

	IBD (*n* = 42)	Control (*n* = 89)
**Sociodemographic features**		
Age in years (mean (range))	34.6 (16–60)	36.3 (15–71)
Female/male (%)	59.5/40.5	64.0/36.0
White/nonwhite (%)	64.3/35.7	65.1/34.9
Smoking (%)	9.5	1.1
**Vaccine (%)**		
AstraZeneca	47.6	64.0
CoronaVac	19.0	19.2
BioNTech Pfizer	33.4	16.8
**Previous coronavirus infection (%)**	21.4	16.8
**Asymptomatic/mild symptoms (%)**	100	100
**CD (%)**	73.8	-
**UC (%)**	26.2	-
**CD Localization (%)**		
L1 (terminal ileum)	29.0	-
L2 (colon)	25.8	-
L3 (ileocolon)	42.0	-
L4 (upper GI tract)	3.2	-
**UC Extension (%)**		
E1 (Proctosigmoiditis)	18.2	-
E2 (Left colitis)	18.2	-
E3 (Pancolitis)	63.6	-
**CD Behavior (%)**		
B1 (nonstricturing nonpenetrating)	61.3	-
B2 (stricturing)	29.0	-
B3 (penetrating)	9.7	-
P (perianal)	25.8	-
**Therapy (%)**		
Salicylate/none	19.0	-
Biologic only	16.7	-
Thiopurine only	31.0	-
Combotherapy	33.3	-

IBD, inflammatory bowel disease; CD, Crohn’s disease; UC, ulcerative colitis.

**Table 2 healthcare-11-02767-t002:** Comparative analysis of vaccination schemes in cases and controls.

Vaccine Scheme	Descriptive Statistics	Control				IBD	
		Pre-Vaccination	One Month after 2nd Dose	*p*	Pre-Vaccination	One Month after 2nd Dose	*p*
**CV**	N	17	17	<0.0001	8	8	0.0078
	Mean	30.9	1397.3		86.8	1140.4	
	Median	6.8	1255.5		48.6	1240.2	
	SD	64.6	854.7		111.4	890.8	
	Min	6.8	315.5		6.8	91.6	
	Max	245.8	3742.5		327.8	2669.7	
**AZ**	N	57	57	<0.0001	20	20	<0.0001
	Mean	83.7	4498.3		147.3	4017.3	
	Median	8.3	2224.9		7.6	1979.5	
	SD	214.0	6085.2		255.5	5032.5	
	Min	4.7	143.0		6.8	100.0	
	Max	1229.4	32,767.2		938.9	21,387.0	
**Pfizer**	N	15	15	<0.0001	14	14	<0.0001
	Mean	200.8	25,648.1		1096.0	19,313.3	
	Median	6.8	23,346.6		10.7	12,061.3	
	SD	554.1	13,650.9		3594.7	17,537.2	
	Min	6.8	10,493.3		0.1	433.0	
	Max	2144.4	58,875.8		13521.5	57,196.8	

AZ, AstraZeneca; CV, CoronaVac; SD, standard deviation; Min, minimum; Max, maximum.

**Table 3 healthcare-11-02767-t003:** Factors associated with the antibody titers against SARS-CoV-2.

Coefficients ^a^							
Variable	Non-Standardized Coefficient		Standardized Coefficient	t	*p* Value	Confidence Interval of 95.0% for B	
	B	Standard Model	Beta			Lower Bound	Upper Bound
(Constant)	1099.12	4241.01		0.259	0.796	−7289.42	9487.67
Case–control	−2103.51	1719.57	−0.089	−1.223	0.223	−550.47	1297.72
Vaccination scheme	5986.88	1162.35	0.409	5.151	0.000	3687.80	8285.97
Previous COVID-19	5356.14	20,285.72	0.185	2.568	0.011	1230.67	9481.62
Age	−182.38	71.03	−0.201	−2.568	0.011	−322.87	−41.89

^a^ Dependent variable: delta_pre_1month.

## Data Availability

Protocols, analytical methods, and study materials are available upon request from interested researchers. The raw data supporting the conclusions of this manuscript will be made available by the authors without undue reservation by any qualified researcher.

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
