# Peer review of "Serum Anti-Spike Antibodies Are Not Affected by Immunosuppressants in SARS-CoV-2 Vaccinations Given to Brazilian Patients with Inflammatory Bowel Disease"

_healthcare, 2023, doi:10.3390/healthcare11202767_

Round 1

Reviewer 1 Report

Good study and well written.

Just a few questions:

1. In table 1, why only females were accounted for? What about the male candidates?

2. The way "Age" is highlighted in bold and then "female, smoking, and white" is listed below, gives an impression as they are subgroups of age.

Line 180-181:

3. "In patients with IBD, the mean antibody titers were 452.0 AU/mL before vaccination and 8568.0 AU/mL after the two vaccine doses."

4. While the IBD patients’ baseline is almost 5 times higher, the authors conclude that the post-vaccination titers are the same in the two groups?

5. Line 318-319, authors discuss the reduced antibody levels with age. It would be better to see the antibody levels in different age groups in a table form, instead of a mean value only. I did not see these values in the tables

Author Response

Reviewer#1

Comments and Suggestions for Authors

Good study and well written.

We thank all three reviewers for their attentive reading our manuscript and for the suggestions that make our work more accurate and clearer.

Just a few questions:

  1. In table 1, why only females were accounted for? What about the male candidates?

Words for clarity.

R: We agree with this reviewer, and we included additional descriptive terms for clarity in Table1.

  1. The way "Age" is highlighted in bold and then "female, smoking, and white" is listed below, gives an impression as they are subgroups of age.

R: We thank this reviewer for his/her attentive reading of our manuscript and we agree with the point raised. We followed his/her suggestion to correct the mistake.

Line 180-181:

  1. "In patients with IBD, the mean antibody titers were 452.0 AU/mL before vaccination and 8568.0 AU/mL after the two vaccine doses."
  2. While the IBD patients’ baseline is almost 5 times higher, the authors conclude that the post-vaccination titers are the same in the two groups?

R: Actually, the mean values presented in the text do not represent the central tendency, due to the skewed distribution, as shown by the individual values in the graph. The graph also shows medians with interquartile ranges, which were more appropriate to indicate the differences detected using the Wilcoxon signed rank test, for pre- versus post-vaccination titers (both in IBD and control groups). Therefore, the titers significantly increased in a similar fashion in both groups, regardless of their baseline values (pre-vaccination).

  1. Line 318-319, authors discuss the reduced antibody levels with age. It would be better to see the antibody levels in different age groups in a table form, instead of a mean value only. I did not see these values in the tables

R: We understand this reviewer’s concern and we prepared an additional Figure (new Supplementary Figure S2), including median values with interquartile ranges, and individual dots, according to age ranges. We did not show the individual values before, because the differences were not significant when analyzing IBD and controls separately. However, when considered together (IBD and controls), as in the model proposed, the reduction in the antibody levels with age becomes significant (statistical significance was achieved by the higher number of individuals in the test).

Reviewer 2 Report

1.       Introduction:

- this paragraph is too long, the authors intention was probably to justify the study but it can be done in less words.

- It would be good to underline the aim of the study clearly in the last paragraph of the introduction.

2.       Methods:

a)       Study design:

-          It is not clear what where the exclusion criteria, in lines 101-107 you state that patients with prior SARS-CV-2 infection were excluded from the study but in the analysis and results you present patients that had the infection prior to first sampling. This brings up some concerns about the design of the study. Please clarify that, either do not analyze those patients or change the exclusion criteria,

-          It would more clear if you would present a graph were a reader can fallow the path of recruitment, and drop outs.

b)      Study protocol and procedures:

-          The appendix is hard to read, looks like raw data put in,

3.       Results:

-          Again the things you mention in lines 154 -159 would be clearer do read if they were presented in a graph,

-          In Table 1 you present results according to therapy and you analyze patients on salicylate or none  but those drugs are not immunosuppressants so the question is were those patients’ on immunosuppressants with salicylate – in that case it is not clear from your table and you have to change it, or you analyzed patients without immunosuppressants- in that case they should be excluded from the analysis.

-          Again in section 3.1 you present results regarding patients with prior infection but according to your exclusion criteria those patients should not be  analyzed.

-          In section 3.2 you presents results regarding the level of antibodies prior vaccination and they are significantly higher in IBD patients, why do you think that is? You do not discuss this later on in the discussion section, do you thing that it influences the results of levels of antibodies after vaccination? Please clarify that.

-          The Figure 2 is hard to read (font is to small).

4.       Discussion:

-          In first paragraph- lines 259-265 you repeat what you said in the introduction in my opinion it not necessary

-          Again in lines 277-282 you repeat the aim of the study which was made in introduction,

-          Lines 337-342- is interesting but not relevant to your study,

5.       Conclusion:

-          In line 362 you state that results of your study support the safety of the vaccination but you did not address this issue in your study, no results about any adverse events in either of examined groups were presented so this statement has no please in your conclusion,

The quality of English is fine.

Author Response

Reviewer #2

Comments and Suggestions for Authors

  1. Introduction:

- this paragraph is too long, the authors intention was probably to justify the study but it can be done in less words.

- It would be good to underline the aim of the study clearly in the last paragraph of the introduction.

R: We agree with this reviewer and we managed to shorten the Introduction as suggested. However, although we also agree that underlining the aim of the study could be interesting, we will leave this decision to the journal’s Editor.

  1. Methods:
  2. a) Study design:

-          It is not clear what where the exclusion criteria, in lines 101-107 you state that patients with prior SARS-CV-2 no infection were excluded from the study but in the analysis and results you present patients that had the infection prior to first sampling. This brings up some concerns about the design of the study. Please clarify that, either do not analyze those patients or change the exclusion criteria,

R: We agree with this reviewer’s comment, and we removed “patients with prior SARS-CoV-2 infection” from the sentence, for correctness. 

-          It would more clear if you would present a graph were a reader can follow the path of recruitment, and drop outs.

R: We thank this reviewer’s suggestion, and we built a flowchart explaining the path followed by the patients in the selection process. We suggest it should appear as a Supplementary Figure S1.

  1. b) Study protocol and procedures:

-          The appendix is hard to read, looks like raw data put in,

R: We understand this reviewer’s difficulty, and the file actually represents raw data. That file is not necessary at the moment and may be presented in response to requests from other researchers.

  1. Results:

-          Again the things you mention in lines 154 -159 would be clearer do read if they were presented in a graph,

R: We understand this reviewer refers to the first paragraph in the Results section. We hope the new supplementary Figure will help to clarify this issue.

-          In Table 1 you present results according to therapy and you analyze patients on salicylate or none  but those drugs are not immunosuppressants so the question is were those patients’ on immunosuppressants with salicylate – in that case it is not clear from your table and you have to change it, or you analyzed patients without immunosuppressants- in that case they should be excluded from the analysis.

R: In Table 1 we described the characteristics of all the patients in the study. We analyzed all patients with inflammatory bowel disease, regardless of the type of treatment employed. Some were using immunosuppressants, while others were not (such as those using only salicylate or those not on any medication at that time). Thus, our aim was to examine not only the outcomes of vaccination in immunosuppressed patients, but also to compare them with those who were not immunosuppressed within the same group of IBD patients. A specific analysis regarding the therapeutic regimen is presented on item 3.5 of Results, “Potential Association between Antibody Titers and the Therapeutic Regimen for IBD”.

-          Again in section 3.1 you present results regarding patients with prior infection but according to your exclusion criteria those patients should not be analyzed.

R: We agree with this comment, and we apologize for the misunderstanding. The problem was solved by removing the mistaken information concerning prior SARS-CoV-2 infection in the Methods section. 

-          In section 3.2 you present results regarding the level of antibodies prior vaccination and they are significantly higher in IBD patients, why do you think that is? You do not discuss this later on in the discussion section, do you thing that it influences the results of levels of antibodies after vaccination? Please clarify that.

R: We understand this reviewer’s concern as the means are considerably different between the groups. However, as shown in Figure 1, the differences are not statistically different (p>0.9999). However, this is explained by the fact that in the IBD group, just a single patient had antibody titers above 17,000.00 AU/ml (likely due to a prior SARS-CoV-2 infection), which raises the mean to higher values. If we consider the median, the values become very similar, being 7.5 AU/ml in the IBD group and 6.8 AU/ml in the control group. In fact, the mean values presented in the text do not represent the central tendency, due to the skewed distribution, as shown by the individual values in the graph. Nevertheless, the graph also shows medians with interquartile ranges, which were more appropriate to indicate the differences detected using the Wilcoxon signed rank test, for pre- versus post-vaccination titers (both in IBD and control groups). Therefore, the titers significantly increased in a similar fashion in both groups, regardless of their baseline values (pre-vaccination).

-          The Figure 2 is hard to read (font is to small).

R: we apologize for the inconvenience, but we tried to follow the journal’s recommendations regarding the article features. We made changes to the figure in order to increase font size.

  1. Discussion:

-          In first paragraph- lines 259-265 you repeat what you said in the introduction in my opinion it not necessary

-          Again in lines 277-282 you repeat the aim of the study which was made in introduction,

-          Lines 337-342- is interesting but not relevant to your study,

R: We agree with these reviewer’s comments, and we removed the unnecessary or repeated information in the text within the Discussion section, as suggested.

  1. Conclusion:

-          In line 362 you state that results of your study support the safety of the vaccination but you did not address this issue in your study, no results about any adverse events in either of examined groups were presented so this statement has no please in your conclusion,  

R: We understand this reviewer’s concern and we agree to remove the statement mentioned by him/her.

Comments on the Quality of English Language

The quality of English is fine.

We thank all three reviewers for their attentive reading our manuscript and for the suggestions that make our work more accurate and clearer.

Reviewer 3 Report

The authors of the paper titled "Humoral Immunity to SARS-CoV-2 Vaccination in Patients With Inflammatory Bowel Disease Is Not Affected by Immunosuppressants" provide a small contribution from Brazil by confirming findings already established by other authors worldwide. This confirmation pertains specifically to seroconversion and SARS-CoV-2 antibody production in IBD patients. The paper is well-written and easy to read, but some questions should be addressed:

1. The Title: The title should be revised since the paper assesses only a portion of humoral immunity, namely the serum anti-spike antibody titers. Furthermore, it's important to note that the cohort studied comprises a small group of Brazilian individuals, which should also be reflected in the title.

2. Line 75: SARS-CoV-2 vaccines are known to reduce the severity of infection in patients but may not necessarily affect transmission.

3. Line 118: The evaluation of antibody titers should be clarified. Specifically, it should be mentioned from which SARS-CoV-2 variant the spike protein was derived in the methods sections. Additionally, discussing the relevance of the selected variant in relation to the type of vaccine used is crucial.

4. If results beyond six months post the second vaccine dose are not presented, it is advisable not to mention them in order to avoid confusion and unnecessary questions for the reader.

5. For figure and table titles, consider using a smaller font size to the text to indicate whether they are part of the results or part of the figure.

6. Line 228: You mentioned a negative association with older age (p < 0.005), but Table 3 displays a p-value of 0.011. Please clarify and provide a consistent value.

7. Line 240: Inform the reader about the specific types of immunosuppressants (IS) that were taken by the IBD patients in the study.

These revisions will help enhance the clarity and accuracy of your text.

Author Response

Reviewer #3

Comments and Suggestions for Authors

The authors of the paper titled "Humoral Immunity to SARS-CoV-2 Vaccination in Patients With Inflammatory Bowel Disease Is Not Affected by Immunosuppressants" provide a small contribution from Brazil by confirming findings already established by other authors worldwide. This confirmation pertains specifically to seroconversion and SARS-CoV-2 antibody production in IBD patients. The paper is well-written and easy to read, but some questions should be addressed:

  1. The Title: The title should be revised since the paper assesses only a portion of humoral immunity, namely the serum anti-spike antibody titers. Furthermore, it's important to note that the cohort studied comprises a small group of Brazilian individuals, which should also be reflected in the titlle.

R: We thank this reviewer for his/her attentive reading of our manuscript, and we followed the suggestion concerning the title. The new title should read: “Serum anti-spike antibodies to SARS-CoV-2 vaccination in Brazilian patients with inflammatory bowel disease are not affected by immunosuppressants“.

  1. Line 75: SARS-CoV-2 vaccines are known to reduce the severity of infection in patients but may not necessarily affect transmission.

R: Currently there are references showing that actually vaccination might reduce transmission. However, we understand this reviewer’s concern, and we changed the sentence following his/her suggestion. “The introduction of SARS-CoV-2 vaccines has led to dramatic reductions in hospitalization and death [15, 16].”

  1. Line 118: The evaluation of antibody titers should be clarified. Specifically, it should be mentioned from which SARS-CoV-2 variant the spike protein was derived in the methods sections. Additionally, discussing the relevance of the selected variant in relation to the type of vaccine used is crucial.

R: We understand this reviewer’s concern. However, in the assay utilized in the study, the spike protein was derived from the wild-type virus, before the emergence of variants. Here, we refer to the study presented in the leaflet of the anti-spike IgG used in this work to explain that this test can also detect variants.

  1. If results beyond six months post the second vaccine dose are not presented, it is advisable not to mention them in order to avoid confusion and unnecessary questions for the reader.

R: Regarding the post-vaccination levels, although in Table 2 we compare titers only before vaccination and one month after the second dose, titers measured 6 months after the second dose are shown in Figures 2 and 4.

  1. For figure and table titles, consider using a smaller font size to the text to indicate whether they are part of the results or part of the figure.

R: We changed font sizes according to this reviewer’s suggestion.

  1. Line 228: You mentioned a negative association with older age (p < 0.005), but Table 3 displays a p-value of 0.011. Please clarify and provide a consistent value.

R: This reviewer’s is right. There was an inconsistency in the text of the Results section. The correct numbers are shown in the Table. Therefore, we made the appropriate corrections in the corresponding text, as indicated by him/her.

  1. Line 240: Inform the reader about the specific types of immunosuppressants (IS) that were taken by the IBD patients in the study.

R: we agree with this comment, and we added explanatory parentheses in the referred paragraph, for clarity.

These revisions will help enhance the clarity and accuracy of your text.

We thank all three reviewers for their attentive reading our manuscript and for the suggestions that make our work more accurate and clearer.